# Long-Term Survival Outcomes beyond the First Year after Liver Transplantation in Pediatric Acute Liver Failure Compared with Biliary Atresia: A Large-Volume Living Donor Liver Transplantation Single-Center Study

**DOI:** 10.3390/jcm11247480

**Published:** 2022-12-16

**Authors:** Sola Lee, Nam-Joon Yi, Eui Soo Han, Su young Hong, Jeong-Moo Lee, Suk Kyun Hong, YoungRok Choi, Hyun-Young Kim, Joong Kee Youn, Dayoung Ko, Jae Sung Ko, Jin Soo Moon, Seong Mi Yang, Kwang-Woong Lee, Kyung-Suk Suh

**Affiliations:** 1Department of Surgery, Seoul National University College of Medicine, Seoul National University Hospital, Seoul 03080, Republic of Korea; 2Department of Pediatrics, Seoul National University College of Medicine, Seoul National University Children’s Hospital, Seoul 03080, Republic of Korea; 3Department of Anesthesiology and Pain Medicine, Seoul National University College of Medicine, Seoul National University Hospital, Seoul 03080, Republic of Korea

**Keywords:** pediatric liver transplantation, acute liver failure, hepatic artery, renal replacement therapy, rejection

## Abstract

Pediatric acute liver failure (PALF) is a common cause of liver transplantation (LT) but showed poor post-LT outcomes. We reviewed 36 PALF patients and 120 BA patients who underwent LT in our institution. The cause of PALF was unknown in 66.7%. PALF patients were older (6.2 vs. 2.9 years) with higher PELD scores (31.5 vs. 24.4) and shorter waitlist time (15.7 vs. 256.1 days) (*p* < 0.01). PALF patients showed higher rates of post-transplant renal replacement therapy (RRT) (13.9% vs. 4.2%) and hepatic artery complications (13.9% vs. 0.8%), while portal vein complications rates were lower (0% vs. 10.8%), (*p* < 0.05). Although PALF patients showed lower 5-year survival rates (77.8% vs. 95.0 %, *p* < 0.01), the 5-year survival rates of patients who lived beyond the first year were comparable (96.6% vs. 98.3%, *p* = 0.516). The most common cause of deaths within one year was graft failure (75.0%) in PALF patients, but infection (67.7%) in BA patients. In multivariate analysis, lower body weight, hepatic artery complications and post-transplant RRT were associated with worse survival outcomes (*p <* 0.05). In conclusion, physicians should be alert to monitor the immediate postoperative graft dysfunction and hepatic artery complications and patients on post-transplant RRT in order to improve survival outcomes in PALF patients.

## 1. Introduction

Pediatric acute liver failure (PALF) is a rapid and progressive clinical syndrome characterized by acute onset of liver injury with coagulopathy [1]. Without liver transplantation (LT), mortality rate can be as high as 50%. Even after LT, survival is only achieved in 50–80% of cases [2,3]. PALF is the second most common cause of pediatric LT in Korea after biliary atresia (BA), which is associated with a 10-year survival rate of 90% [4,5].

The etiology of PALF in Korea is different from that in western countries where a higher incidence of acetaminophen-related acute liver failure, which shows favorable outcomes, has been reported [1]. In Korea, cryptogenic acute hepatitis is the most common cause of PALF [4,5] and is associated with poor outcomes. Patients with PALF of unknown etiology may experience recurrent liver failure, as preventing recurrence of the original disease can be challenging and patients will require retransplantation. 

Many studies investigating PALF outcomes come from deceased donor LT (DDLT)-dominant countries [3,6,7]. Due to the shortage of organs from deceased donors, living donor LT (LDLT) is the dominant approach in many Asian countries, including Korea. Considering these differences, we felt the need for a comparable reference group within the country. 

This study analyzed the long-term outcomes of patients with PALF who underwent LT (PALF group) in a single, large-volume LDLT center and compared these with post-LT outcomes of patients with BA. 

## 2. Materials and Methods

### 2.1. Definition of Pediatric Acute Liver Failure and Initial Evaluation at SNUH

We defined PALF as the acute onset of severe liver injury in children without chronic liver disease and with biochemical evidence of liver injury with coagulopathy (prothrombin time international normalization ratio (PT INR) ≥ 1.5 with hepatic encephalopathy or PT INR ≥ 2 regardless of hepatic encephalopathy) according to the Pediatric Acute Liver Failure Study Group guidelines [1,8]. Acute onset was defined as symptoms developing within 8 weeks according to the Korean Network for Organ Sharing (KONOS) [9,10].

Patients with acute deterioration of liver function are initially admitted to the pediatric department for diagnosis and medical management. After consulting with the LT team and counseling family members, the organ transplantation center lists patients for DDLT. At the same time, the living donor workup process is initiated if there is an available potential donor in the family. Because of organ shortages, only a limited number of patients can undergo DDLT. During the waitlist time, some patients present with sudden clinical deterioration reflecting a later stage of liver failure (i.e., acute worsening of mental status, hypoglycemia), and prompt LT may be necessary. Initial evaluation of the patient includes establishing medical history, physical examination, psychological assessment, blood tests, and computed tomography imaging and echocardiography.

Contraindications to liver transplantation in pediatric patients with PALF include irreversible brain damage, uncontrolled infection, severe cardiopulmonary disease, and some etiologies such as severe multisystem mitochondrial disease and Niemann-Pick disease type C.

### 2.2. Patient Selection

A total of 225 patients aged < 18 years underwent LT at the Seoul National University Hospital between January 2000 and December 2015. In Korea, a pediatric patient is defined as having an age of less than 18 years old [9]. The patient medical records were reviewed and categorized according to the etiology of the underlying liver disease. Thirty-six patients underwent LT for PALF and 120 patients underwent LT for BA. Sixty-nine patients were excluded from the study because they had undergone retransplantation or primary LT for etiologies other than PALF or BA (Figure 1).

### 2.3. Postoperative Management

Postoperative immunosuppression therapy consisted of tacrolimus and steroids. Steroids were gradually decreased and usually discontinued 6 months after surgery. In cases of renal insufficiency, the tacrolimus dose was minimized and combination therapy with basiliximab and mycophenolate was introduced. Blood tests and Doppler ultrasound of the transplanted liver were performed daily for the first week and twice per week after the first week. Prostagalndin E1, antithrombin III and nafamostat mesilate were administered while in the intensive care unit up to seven days. However, anticoagulants such as heparin or lower molecular weight heparin was not administered routinely. Aspirin was administered once a regular diet was tolerated by patients with low bleeding risk (platelet count > 50,000/mm^3^ and PT INR less than 2). 

### 2.4. Parameters

The following data were collected from a retrospective review of the transplant recipients’ medical records: patient age, sex, body weight, height, etiology of liver failure, pediatric end-stage liver disease (PELD) score, Child-Pugh score, waitlist time before LT, and year of transplantation. Data on donor characteristics included age and relationship with the patient. The following operative and graft factors were also collected: type of transplantation (living vs. deceased donor), graft type (living vs. split liver vs. whole liver), operative time, estimated blood loss, and ABO compatibility. 

Postoperative outcomes included surgical complications (major complications > grade 2 of the Clavien-Dindo classification), infection, mechanical ventilation, renal replacement therapy (RRT), perioperative vasopressor use, number of days in the intensive care unit, total post-transplant hospital stay duration, acute rejection, and patient and graft survival outcomes. Rejection was defined as biopsy proven rejection, which was classified according to the Banff criteria.

Graft failure after transplantation was defined as death or re-transplantation from liver-related complication such as rejection or recurrence of liver failure. Primary non-function was defined as graft dysfunction with at least two of the following: AST ≥ 2000, INR ≥ 2.5, total bilirubin ≥ 10.0 mg/dL, and acidemia (pH ≤ 7.3 or lactate ≥ 4 mMol/L) within 7 days of transplantation [9]. Liver failure as part of multiple-organ failure from septic shock of identified etiology was not considered as graft failure. However, if a patient died or underwent re-transplantation from recurrent biliary sepsis, it was considered as graft failure.

### 2.5. Statistical Analysis

Statistical analyses were performed using the IBM SPSS software package (version 25.0; Armonk, NY, USA). Quantitative data are reported as mean (±standard deviation). Categorical values are expressed as numbers (percentage, %). Categorical values were analyzed using the chi-square test and continuous variables were analyzed using the Mann–Whitney U test. One-year and 5-year survival rates were calculated using the Kaplan–Meier method. Cox proportional hazard regression analysis was used to evaluate factors associated with patient survival. A *p*-value less than 0.05 was considered significant. Multivariate Cox proportional hazard regression with backward selection was used to evaluate factors from the univariate analysis. 

### 2.6. Ethics Statement

This study was approved by the Institutional Review Board of Seoul National University Hospital (Number 2206-110-1333). The board waived the requirement for informed consent due to the retrospective nature of this study. 

## 3. Results

### 3.1. Clinical and Operative Characteristics

The demographic and clinical characteristics of the patients are described in Table 1. Comparison of clinical characteristics between the PALF and BA group showed that patients in the PALF group were older (6.2 vs. 2.9 years old, *p* < 0.01), taller (109 vs. 84 cm, *p* < 0.01), and had a greater body weight (25.7 vs. 13.8 kg, *p* < 0.01). The PALF group had higher PELD scores (31.5 vs. 24.4, *p* < 0.01) and shorter waitlist times (15.7 vs. 256.1 days, *p* < 0.01) than did the BA group, reflecting the increased severity of the disease. However, graft types were not different between the two groups; LDLT was dominant and was present in more than two thirds in both groups (77.8 vs. 68.3%, *p* = 0.276). Operation duration was shorter in the PALF group than in the BA group (387.6 vs. 424.1 min, *p* < 0.044). There were no differences between the two groups in the sex distribution of the respective donors, although donors donating to patients with PALF were older (34.6 vs. 28.1 years old, *p* < 0.01). In most cases of PALF, the underlying disease was unknown (*n* = 24, 66.7%). In the remaining patients, the following causes were identified: Wilson’s disease (*n* = 5), gestational alloimmune liver disease (*n* = 2), tyrosinemia (*n* = 1), acute hepatitis B infection (*n* = 1), lymphoma (*n* = 1), glycogen storage disease (*n* = 1), and drug-induced liver failure (dapsone, *n* = 1). 

The median follow-up period was 133 months (range, 0.48–225 months). The BA group had a longer follow-up period (158 months vs. 104 months, *p* < 0.01). A higher proportion of patients with BA underwent LT during 2000–2010 (*p* = 0.044).

### 3.2. Postoperative Outcomes

The postoperative outcomes are summarized in Table 2. There were no significant differences in the rates of vasopressor use, ventilator application, length of intensive care unit stays, and hospital stay between the two groups (*p* > 0.05). PALF patients showed higher rates of post-transplant renal replacement therapy (RRT) (13.9% vs. 4.2%) (*p* = 0.049). Of the five patients who required post-transplant RRT, two patients from the PALF were on pre-transplant RRT. All of the five PALF patients had recovery of graft function. In the BA group, none of the five patients with post-transplant RRT were on pre-transplant RRT. However, one patient experienced primary nonfunction and two patients had early allograft dysfunctions. The former patient died from brain death and pneumonia. One of the patients with early allograft dysfunction underwent retransplantation the 25th postoperative day. 

The incidence of hepatic artery (HA) complications was higher in the PALF group (13.9% vs. 0.8%, *p* < 0.01). In the PALF group, hepatic artery complications included four patients with hepatic artery thrombosis which required surgical revision. In two patients, graft function was recovered after surgical revision, but two patients had graft failure. There was also one patient with hepatic artery stenosis who died from complicated CMV colitis and pneumonitis.

A higher rate of portal vein complications was observed in the BA group (10.8% vs. 0%, *p* = 0.039). Thirteen among 120 BA patients experienced portal vein complications: 8 stenosis and 5 thrombosis. For eight patients with portal vein stenosis, 2 patients underwent surgical intervention and 6 patients underwent radiologic interventions with ballooning (*n* = 4) and stenting (*n* = 2). Among 5 patients with portal vein thrombosis, 2 patients underwent radiologic interventions, and three patients underwent surgical thrombectomy. Other postoperative complications, both surgical and medical, were comparable between the groups (*p* > 0.05). 

### 3.3. Patient Survival Outcomes

One- and five-year patient survival outcomes are shown in Figure 2A. The 1-year survival (80.6% vs. 96.7%, *p* < 0.01) and 5-year survival (77.8% vs. 95.0 %, *p* < 0.01) rates were lower in the PALF group than in the BA group. Most deaths occurred in the first year in the PALF group and there was no significant difference between the groups in the 5-year survival outcomes of patients who lived beyond the first year after transplantation (96.6% vs. 98.3%, *p* = 0.516) (Figure 3). 

The most common cause of death in the BA group was infection (*n* = 4, 66.7%), followed by graft failure (*n* = 2, 33.3%) (Table 2). In the PALF group, graft failure (*n* = 6, 75.0%) was the most common cause of death followed by infection (*n* = 2, 25.0%). 

### 3.4. Graft Survival Outcomes

Seven (5.8%) patients with BA had graft failure, while 7 (19.4%) patients with PALF had graft failure (Figure 2B). Of those with graft failure in the BA group, six (85.7%) underwent retransplantation, and of these, five patients survived (71.4%). However, three patients (42.9%) with graft failure in the PALF group underwent retransplantation and, of these, only one survived (14.3%). Rates of graft failure were higher in the PALF group (19.4% vs. 5.8% in the BA group, *p* = 0.02), and survival outcomes of patients with graft failure were worse, with death occurring in 6 out of 7 patients in the PALF group and 2 out of 7 in the BA group (*p* = 0.103). 

The details of graft failure in patients with PALF are summarized in Table 3. Among patients with graft failure resulting in death, two patients (#1, 2) had graft dysfunction of unknown origin. Two patients (#3 and #4) died of graft failure related to biopsy-proven acute cellular rejection, which did not respond to treatment. One patient (#5), who received an ABO-incompatible living donor graft from her mother, underwent retransplantation and the explanted graft showed massive hepatic necrosis with C4d deposition. One patient (#6) underwent LDLT for cryptogenic PALF and recovered well. However, the graft progressed to liver failure again due to a possible recurrence of the primary disease 3 months after initial transplantation. After retransplantation, the patient died of graft-versus-host disease. The last patient (#7) experienced graft failure after primary LDLT for cryptogenic PALF because of HA thrombosis and intrahepatic biliary strictures. The patient underwent retransplantation from a deceased donor for recurrent biliary sepsis.

### 3.5. Risk Factors Associated with Patient Survival

Risk factors associated with patient survival are described in Table 4. Univariate analysis showed that body weight (OR 0.199, CI 0.055–0.715, *p* = 0.013), PALF etiology (OR 5.06, CI 1.753–14.606, *p* < 0.01), HA complications (OR 9.555, CI 2.650–34.452, *p* < 0.01), and need for postoperative RRT (OR 6.038, CI 1.1891–19.285, *p* < 0.01) were associated with survival outcomes. In the multivariate analysis, the etiology was not significant. Body weight less than 6 kg (OR 0.14, CI 0.035–0.565, *p* < 0.01), HA complications (OR 10.264, CI 2.310–45.610, *p* < 0.01), and postoperative RRT (OR 9.318, CI 2.363–36.748, *p* < 0.01) were associated with worse survival outcomes.

## 4. Discussion

### 4.1. Summary & Comparative Recent Works

We analyzed the post-transplant long-term outcomes of pediatric patients with acute liver failure and compared them to the survival of patients with BA at a single LDLT-dominant center. As expected, the 5-year survival rate in the PALF group was significantly lower than that in the BA group. Since most deaths occurred during the first posttransplant year, we analyzed survival rates of patients who survived beyond the first year. We found that there was no difference in survival rates between the two groups. In our study, the BA group had a higher proportion of DDLT cases (31.7%), of which two thirds were split-liver transplants, compared to the PALF group (20.8%), although this was not statistically different. The average waitlist time for patients with BA who underwent DDLT was 256 days. However, patients with PALF required transplantation more urgently; therefore, LDLT was performed more frequently in this group. Subgroup analysis of patients who underwent LDLT revealed that survival outcomes were also significantly lower in the PALF group than in the BA group (71.4% vs. 93.9% in BA, *p* < 0.01). This finding indicated that LDLT was not related to survival outcomes. The lower survival rates observed in the PALF group may be explained by the higher rates of HA complications and post-transplant RRT due to serious pre-transplant condition. If PALF recipients overcome and survive the immediate post-transplant period, the survival outcomes are similar to the outcomes of patients with BA. 

### 4.2. Prognostic Criteria and Decision for LT

Many prognostic criteria for liver failure have been proposed [10,11,12]. The King’s College criteria and the Clichy criteria, which are used to decide whether patients with liver failure should undergo LT, are based on etiology, biochemical change, and the presence of hepatic encephalopathy. However, these criteria cannot reliably predict outcomes in pediatric patients. In children, mental status examination is often difficult, and hepatic encephalopathy may not be apparent until the later stages of liver failure. At our center, the decision to proceed with LT is based on the trend in the biochemical changes with emphasis on clinical deterioration, as proposed by Lee et al. [8]. However, as the present study retrospectively examined survival outcomes after transplantation, we did not use Lee’s transplantation decision making tree.

### 4.3. Factors Associated with Mortality and Hepatic Artery Complications

Through multivariate analyses, we identified HA complications, the need for post-transplantation RRT, and body weight less than 6 kg as the risk factors associated with poor prognosis. Pediatric LT patients are more prone to HA complications because of the smaller size of their hepatic arteries. Even with successful reconstitution of hepatic arterial flow after thrombosis, patients experienced higher rates of biliary complications, which were serious, leading to graft loss [13,14]. Our data showed that although average age and body weight were higher in the PALF group, HA complications were significantly higher. This finding may be explained by the differences in the underlying disease mechanisms. PALF affects previously healthy individuals without portal hypertension, so there is no arterial hyperplasia caused by arterial hyperperfusion secondary to portal hypertension. In contrast, patients with BA are prone to portal vein hypoplasia and portal hypertension with a relatively enlarged hepatic artery size and flow [15,16,17]. For this reason, the PALF group showed more HA complications, increasing the risk of graft loss and mortality. As discussed in graft failure section, most graft failures were not directly related to hepatic artery complications except in one patient. However, any second hit like acute cellular rejection or sepsis to the patient with HA complication may lead the graft failure or patient death. In addition, the number of mortality cases was small (*n* = 14) in this cohort. For those reasons, when we combined two groups for risk factor analysis, we found that HA complication was associated with increased risk of mortality. Thus, we should more closely monitor patients who had HA complication than other patients regardless of their age and consider an additional anticoagulant protocol in PALF recipients. 

### 4.4. Determining the Etiology of PALF and the Importance of Age Specific Diagnostic Test

Two thirds (66.7%) of causes of PALF were cryptogenic in this study. Determining the etiology of PALF is important because in some patients with cryptogenic PALF, who may have had spontaneous recovery or may experience recurrent liver failure even after LT, transplantation may be futile [18]. In addition to medical support, initial management should include thorough evaluation of the etiology of PALF and a multidisciplinary decision-making process to determine whether to proceed with LT or not. This involves taking a history of recent medications, illness, and underlying disease, and blood and urine testing to detect viral hepatitis, autoimmune hepatitis, genetic disorders, and immunologic syndromes such as hemophagocytic lymphohistiocytosis. However, complete evaluation is not always possible, and other centers have reported that the complete workup rate is approximately 50% before a decision to transplant is made [1,19]. Moreover, in this study, seven patients with PALF with graft failure underwent testing for the following etiologies before transplantation: hepatitis A, hepatitis B, hepatitis C, CMV, Epstein–Barr virus, toxoplasma, rubella, herpes simplex virus, autoimmune hepatitis, Wilson’s disease, hemophagocytic lymphohistiocytosis and gestational alloimmune liver disease according to their age. Age-specific diagnostic testing was proposed by Narkewicz et al., based on the frequency and characteristics of age-related liver failure in pediatric patients [18]. These tests were recommended for a complete but not exhaustive evaluation. 

### 4.5. Mortality and Causes of Graft Failure in PALF Group

Although the survival outcome of the PALF group was worse than that of the BA group, underlying liver disease itself was not a risk factor of overall survival outcomes in multivariate analysis. In the analysis of risk factors associated with 1-year mortality, PALF remained as a significant risk factor (OR 4.285, CI 1.092–16.812, *p* = 0.037). However, the power was lost in the long-term follow-up, because for patients who survived the first posttransplant year, there was no difference in survival between two groups. Although we performed LDLTs around 2 weeks in the PALF group (15.7 days), graft types and waiting times were not associated with survival outcomes. The most common cause of deaths was graft failures which did not respond well to treatment including retransplantation. In the PALF group (Table 3), initial graft functions were not well recovered related to delayed graft recovery, primary non-function or acute rejection which did not respond to the usual steroid pulse therapy. In one patient (#5) who died during retransplantation from graft failure due to antibody mediated rejection, the primary LT was an ABO-incompatible liver from the patient’s mother. The patient was a 3-month-old baby with a preoperative IgG-anti B titer of 1:2, and negative donor-specific antibody (flow cytometry test was negative for both T- and B-cells, and the PRA panel was negative). Therefore, the desensitization protocol was not applied following our center’s protocol. However, the explanted liver showed massive hepatic necrosis related to antibody-mediated rejection. Immune function may play different roles in posttransplant graft survival in the two groups. In the BA group, the graft rejection rates were not significantly different from those in the PALF group. However, BA recipients responded well to the usual graft rejection treatment. In contrast, rejection leads to graft failure and patient death in previously healthy PALF recipients. A more competent immune function of PALF recipients with poor treatment response to rejection therapy may be important factors in rejection-related graft failure in the PALF group. Considering these situations, evaluation of graft dysfunction should include graft biopsy (which is the only way to differentiate graft rejection from other causes) and imaging studies (used to screen vascular and biliary complications) in PALF recipients. We did not evaluate immune function in this cohort because of the retrospective nature of data collection. Further studies should be planned to reveal the mechanism of severe graft failure in PALF recipients. 

Among the seven patients who underwent first transplantation due to PALF and lost the graft, three (#1, #2, and #6) patients suffered from graft failure without definite cause; however, the onsets were different (from one week to three months after transplantation). Risk factors associated with primary non-function vary according to donor factors (donor age, steatosis), prolonged cold ischemic time, and recipient factors (use of high-dose vasopressors and life support) [20,21]. However, in these cases, the donors were young, there was a very short cold ischemic time, and there were minimal fatty changes in the living donors. Patient #6 showed severe graft dysfunction at an outpatient clinic 3 months after transplantation. The explanted graft in retransplantation showed massive hepatic necrosis. The other four patients experienced acute rejection which did not respond well to treatment. Therefore, physicians should pay attention to graft dysfunction of unknown origin in PALF, which can potentially be related to primary disease recurrence or severe rejection, particularly in cryptogenic PALF. When we compared survival outcomes among patients with BA, PALF due to other causes, and cryptogenic PALF, the survival outcomes were significantly worse in patients with cryptogenic PALF (*p* < 0.01) (Figure 4). 

### 4.6. Strengths and Limitations of This Study

A strength of this study was the unique Asian organ transplant context, in which living donation is dominant. Overall, 78% of patients with PALF who required LT underwent LDLTs, although it is important to note that the use of a living donor itself was not a better prognostic factor in this cohort. Although the survival outcomes were worse in the PALF group than those in the BA group, the 5-year survival rate was 77.8%, which was comparable to other studies [22,23,24,25]. A further strength was that detailed data analysis could be performed because this was a single center study. However, the limitation of this study was its retrospective nature and small sample size. For this reason, we did not perform a matched comparison between the two groups. 

## 5. Conclusions

In conclusion, although the overall survival and graft survival outcomes of the PALF group were worse than those of the BA group, there was no difference in survival between the two groups beyond the first year post-LT. Graft failure-related deaths rates were higher in the PALF group within one year after LT. Lower body weight, HA complications, and the need for post-transplant RRT were identified as poor prognostic factors in all. A greater focus on immediate postoperative care is required, which includes monitoring for graft dysfunction and HA complications and paying special attention in cases of post-transplant RRT in order to improve one-year survival outcomes, especially in PALF recipients. 

## Figures and Tables

**Figure 1 jcm-11-07480-f001:**
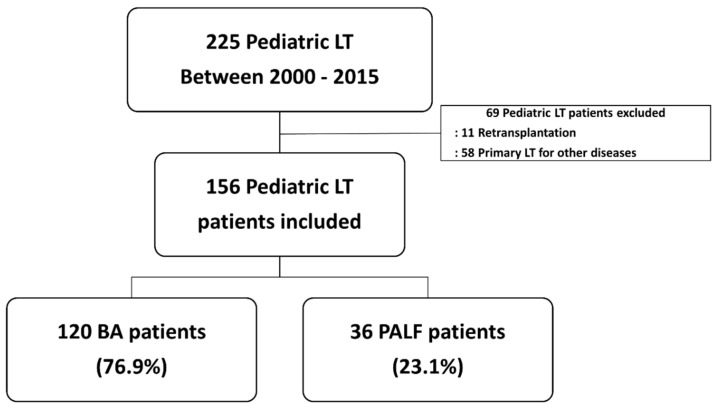
Flow chart of patient selection. Abbreviations: BA, biliary atresia; LT, liver transplantation; PALF, pediatric acute liver failure.

**Figure 2 jcm-11-07480-f002:**
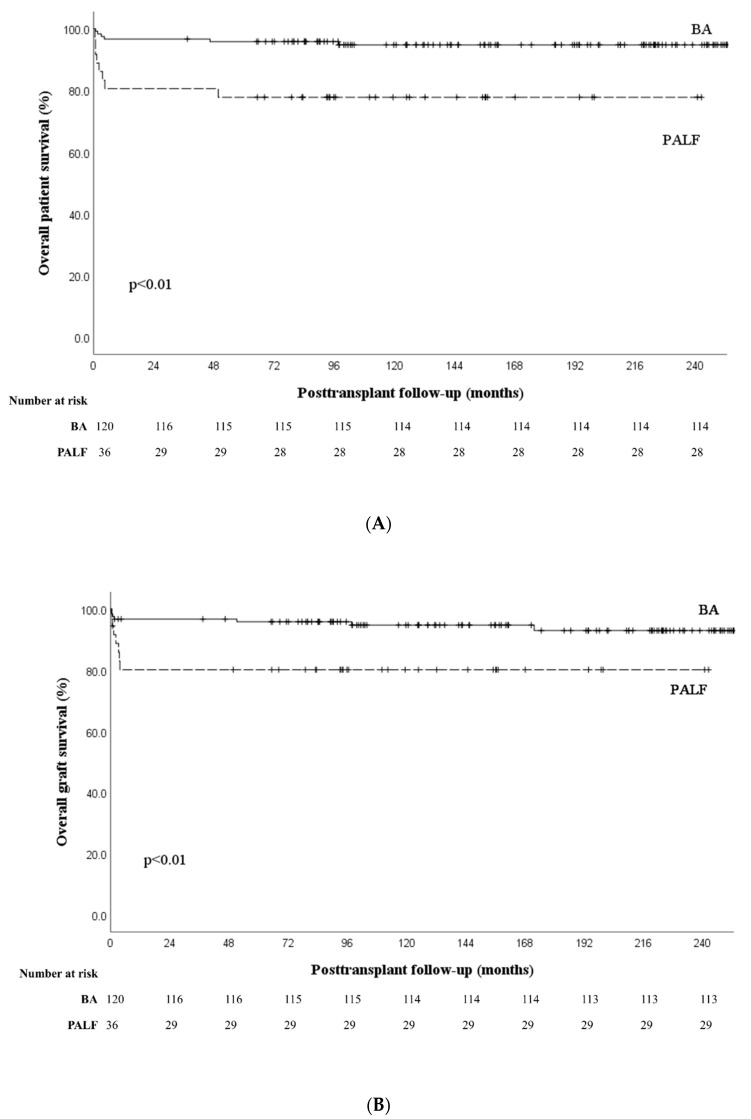
Survival graphs comparing biliary atresia and pediatric acute liver failure groups. (**A**) Patient survival rates. (**B**) Graft survival rates. Abbreviations: BA, biliary atresia; PALF, pediatric acute liver failure.

**Figure 3 jcm-11-07480-f003:**
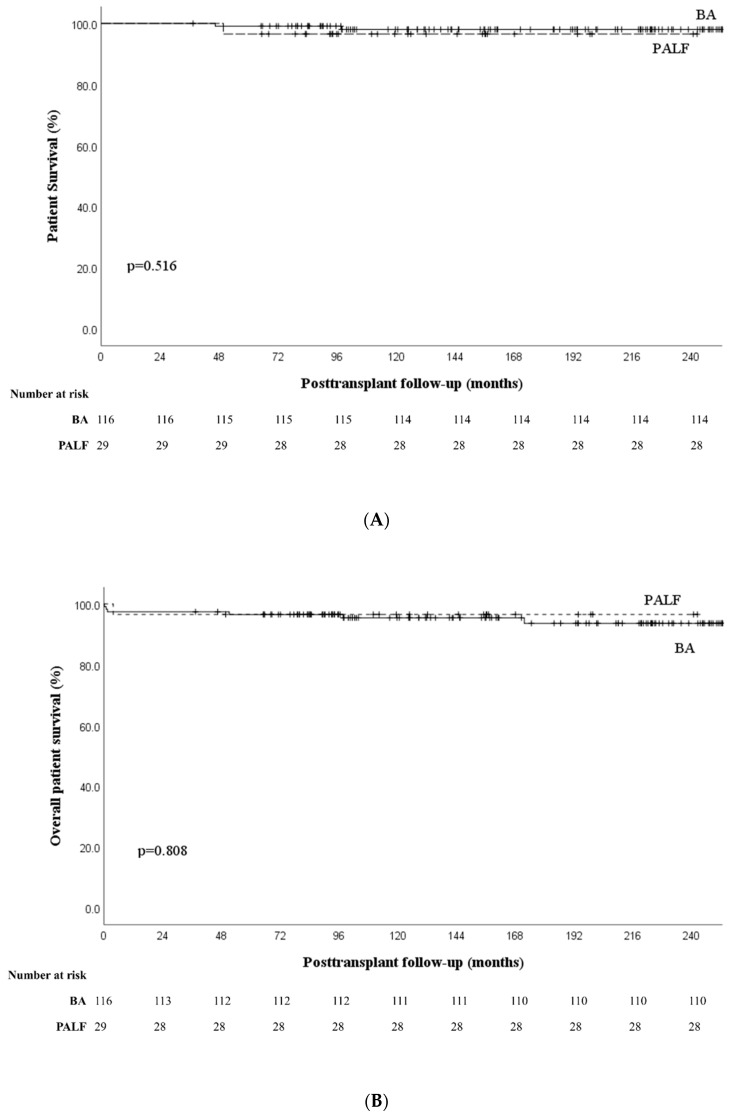
Survival rates in patients living beyond the first post-transplant year, comparing biliary atresia and pediatric acute liver failure groups. (**A**) Patient survival rates. (**B**) Graft survival rates. Abbreviations: BA, biliary atresia; PALF, pediatric acute liver failure.

**Figure 4 jcm-11-07480-f004:**
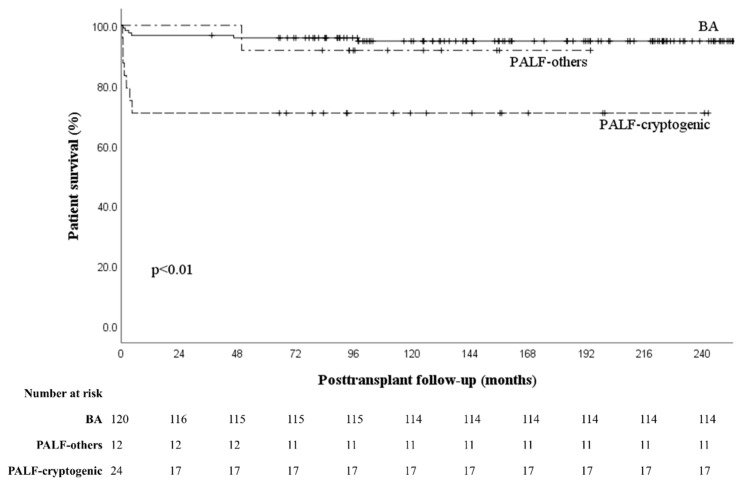
Comparison of patient survival outcomes (biliary atresia vs. cryptogenic pediatric acute liver failure vs. other causes of pediatric acute liver failure). Abbreviations: BA, biliary atresia; PALF, pediatric acute liver failure.

**Table 1 jcm-11-07480-t001:** Clinical and operative characteristics.

Variables	BA (*n* = 120)	PALF (*n* = 36)	*p*-Value
Patient characteristics			
Sex, male:female (%)	45 (37.5):75 (62.5)	19 (52.8%):17 (47.2)	0.102
Age (year)	2.9 ± 3.6	6.2 ± 5.9	<0.01
Body weight (kg)	13.8 ± 10.8	25.7 ± 20.1	<0.01
Height (cm)	84.7 ± 29.9	109.1 ± 44.6	<0.01
PELD score	24.4 ± 14.6	31.5 ± 14.0	<0.01
Child-Pugh score	8.6 ± 2.0	9.9 ± 1.5	<0.01
Waitlist time (days)	256.1 ± 495.2	15.7 ± 29.3	<0.01
Transplant year (%)			
2000–2010:2011–2015	82 (68.3):38 (31.7)	18 (50.0):18 (50.0)	0.044
Follow-up period (months, median)	158.5 ± 69.2 (158.0)	104.1 ± 68.4 (99.1)	<0.01
Operative characteristics			
Donor age (year)	28.1 ± 11.4	34.6 ± 9.4	<0.01
Donor sex, male:female	68 (56.7):52 (43.3)	17 (47.2):19 (52.8)	0.318
LDLT:DDLT, *n* (%)	82 (68.3):38 (31.7)	28 (77.8):8 (22.2)	0.276
Graft type, *n* (%)			0.225
LDLT	82 (68.3)	28 (77.8)	
Split DDLT	25 (20.8)	3 (8.3)	
Whole DDLT	13 (10.8)	5 (13.9)	
ABO incompatibility, *n* (%)	3 (2.5)	3 (8.3)	0.136
Operative time (min)	424.1 ± 98.4	387.6 ± 65.9	0.043
Estimated blood loss (mL)	628.4 ± 1014.5	714.8 ± 570.3	0.681

Abbreviations: BA, biliary atresia; LDLT, living donor liver transplantation; DDLT, deceased donor liver transplantation; PALF, pediatric acute liver failure; PELD, pediatric end-stage liver disease.

**Table 2 jcm-11-07480-t002:** Comparison of postoperative outcomes.

Variables	BA (*n* = 120)		PALF (*n* = 36)		*p*-Value
Immediate postoperative course					
Postoperative ICU stay (days)	11.96 ± 15.2		9.1 ± 6.4		0.271
Postoperative ventilator *, *n* (%)	102 (85.0)		29 (80.6)		0.524
Postoperative RRT, *n* (%)	5 (4.2)		5 (13.9)		0.049
Postoperative vasopressor use, *n* (%)	53 (44.2)		17 (47.2)		0.746
Postoperative hospital stay (days)	34.3 ± 18.7		33.3 ± 29.0		0.816
Surgical complication, *n* (%)					
Hepatic vein	10 (8.3)		5 (13.9)		0.339
Hepatic artery	1 (0.8)		5 (13.9)		<0.01
Portal vein	13 (10.8)		0 (0)		0.040
Bile duct	8 (6.7)		5 (13.9)		0.179
Medical complication, *n* (%)					
Infections					
Viral	29 (24.2)		8 (22.2)		0.810
Bacterial	22 (18.3)		5 (13.9)		0.536
Fungal	1 (0.8)		1(2.8)		0.409
Rejection	15 (12.5)		8 (22.2)		0.149
PTLD	13 (10.8)		2 (5.6)		0.279
Mortality, *n* (%)	6 (5.0)		8 (22.2)		<0.01
Cause of death	1-yr	5-yr	1-yr	5-yr	
Graft failure	1 (25.0)	1 (50.0)	6 (85.7)	0	
Infection	3 (75.0)	1 (50.0)	1 (14.3)	1 (100)	
PTLD	0	0	0	0	

* More than one day after transplantation. Abbreviations: BA, biliary atresia; ICU, intensive care unit; PALF, pediatric acute liver failure; PTLD, post-transplant lymphoproliferative disease; PELD, pediatric end-stage liver disease; RRT, renal replacement therapy.

**Table 3 jcm-11-07480-t003:** Case details of graft failure in patients with PALF.

No.	Age (Year)/Sex	PELD Score	Etiology	Donor Type/Age(Year)/Relation	ABO Incompatibility	Surgical Complication			RRT	Rejection (RAI Score)	Cause of Graft Failure/Death	Re-LT	Survival (Months)
						HA	HV	PV					
1	6/M	43	Cryptogenic	LD/25/ mother	-	-	-	-	-	-	Delayed graft dysfunction/ Dead due to brain death	-	1.2
2	4/F	28	Cryptogenic	LD/39/ father	-	-	-	-	-	-	Primary nonfunction/ Dead	-	0.23
3	0.4/F	44	Cryptogenic	LD/28/ mother	-	+	-	-	-	ACR (5, 8)	ACR/ Dead	-	3.5
4	1.3/M	48	Cryptogenic	LD/30/ unrelated	-	-	-	-	-	ACR (3, 4)	ACR/ Dead	-	0.7
5	0.3/F	13	Cryptogenic	LD/37/ mother	+	-	+	-	-	AMR, hepatic necrosis	AMR/ Dead	+(LD)	2.1
6	2/M	17	Cryptogenic	LD/49/ father	-	-	+	-	+	-	Recurrence of primary disease/ Dead due to GVHD after re-LT	+(DD)	4.4
7	0.8/M	10	Cryptogenic	LD/36/ mother	-	+	-	-	-	ACR (5)	HA thrombosis/ Alive after re-LT	+(DD)	45.0

Abbreviations: PALF, pediatric acute liver failure; ACR: acute cellular rejection, AMR: antibody-mediated rejection, C: cryptogenic, CMV: cytomegalovirus, F: female, GVHD: graft-versus-host disease, HA: hepatic artery, HV: hepatic vein, LD: living donor, M: male, PV: portal vein, RAI: rejection activity index, RRT: renal replacement therapy.

**Table 4 jcm-11-07480-t004:** Risk factors associated with patient survival outcomes.

Variables	Univariate			Multivariate		
	OR	95% CI	*p*	OR	95% CI	*p*
Sex						
Male	Ref					
Female	0.925	0.321–2.665	0.885			
Age (yr)						
<6 months	Ref					
≥6 months	0.590	0.132–2.646	0.491			
Body weight						
<6 kg	Ref			Ref		
≥6 kg	0.199	0.055–0.715	0.013	0.140	0.035–0.565	<0.01
PELD score						
<30	Ref					
≥30	1.784	0.625–5.087	0.279			
Child Pugh score						
<8	Ref					
≥8	2.663	0.743–9.550	0.133			
Waiting time						
<7 days	Ref					
≥7 days	0.565	0.177–1.803	0.335			
Etiology						
BA	Ref			Ref		
PALF	5.06	1.753–14.606	<0.01	2.032	0.647–6.381	0.225
Year of transplant						
2000–2010	Ref					
2011–2015	0.744	0.232–2.386	0.619			
Donor age						
<18 y	Ref					
≥18 y	1.844	0.241–14.098	0.555			
Types of LT						
LDLT	Ref					
DDLT	0.179	0.023–1.369	0.097			
Operative time						
<360 min	Ref					
≥360 min	0.903	0.283–2.880	0.863			
Postoperative RRT						
RRT (−)	Ref			Ref		
RRT (+)	6.038	1.891–19.285	<0.01	9.318	2.363–36.748	<0.01
ICU stay (days)						
<5 days	Ref					
≥5 days	4.908	0.642–37.525	0.125			
Hospital stay (days)						
<30 days	Ref					
≥30 days	0.921	0.319–2.654	0.879			
HA complication						
(−)	Ref			Ref		
(+)	9.555	2.650–34.452	<0.01	10.264	2.310–45.610	<0.01
PV complication						
(−)	Ref					
(+)	0.799	0.104–6.106	0.828			
Infection						
(−)	Ref					
(+)	0.236	0.053–1.053	0.058			
PTLD						
(−)	Ref					
(+)	0.679	0.089–5.194	0.709			
Rejection						
(−)	Ref					
(+)	0.972	0.218–4.344	0.970			

Abbreviations: BA, biliary atresia; DDLT, deceased donor liver transplantation; ICU, intensive care unit; HA, hepatic artery; LDLT, living donor liver transplantation; PALF, pediatric acute liver failure; PELD, pediatric end-stage liver disease; PTLD, post-transplantation lymphoproliferative disorder; PV, portal vein; RRT, renal replacement therapy; OR, odds ratio; CI, confidence interval.

## Data Availability

Data sharing is not applicable to this article.

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
