# Peer review of "Long-Term Survival Outcomes beyond the First Year after Liver Transplantation in Pediatric Acute Liver Failure Compared with Biliary Atresia: A Large-Volume Living Donor Liver Transplantation Single-Center Study"

_jcm, 2022, doi:10.3390/jcm11247480_

Round 1
Reviewer 1 Report
The authors have compared outcomes of pediatric liver transplantation for acute liver failure and biliary atresia in a retrospective study and found that the former has inferior short term (1 year) graft and patient survival as compared to the latter. Congratulations on overall good outcomes, even in the sick cohort with PALF.
Does the term hepatic artery complication in this manuscript mean hepatic artery thrombosis?
Details of the temporal course, outcome of 5 patients with hepatic artery complications in the PALF group are not clear. Two of them had graft failure as per Table 3. In one the cause of death was acute cellular rejection; the other patient was salvaged with a retransplant. What about the other 3 patients?
On multivariable analysis, hepatic artery complications are a significant contributor to mortality. On reading the manuscript however, it seems like other causes resulted in graft and patient loss in the PALF group. The authors could perhaps refine the discussion a little for more clarity to the readers on this aspect.
Did the all patients with post transplant renal replacement therapy have early allograft dysfunction? Did any of them have primary non function? Was there any pre transplant renal impairment in these patients?
ACR as a cause of death is unusual in liver transplantation. Were all cases of ACR biopsy confirmed?
Understandably patients with BA had a higher incidence of portal vein complications. How were they handled?
Reviewer 2 Report
A retrospective study of long-term survival outcomes of PALF and biliary atresia patients was carried out in this manuscript. The results are excellent by international standards. This study could be one of the data for adding evidence to the importance of delicate immediate postoperative management for Post-LT PALF patients. The purpose of this study is well-designed, but this paper exposes minor issues in the data interpretation.
More specific comments are below.
3.5. Risk factors associated with patient survival
Risk factors associated with patient survival are described in Table 4. Univariate 234 analysis showed that body weight (OR 0.199, CI 0.055-0.715, p =0.013), PALF etiology (OR 235 5.06, CI 1.753-14.606, p<0.01), HA complications (OR 9.555, CI 2.650-34.452, p <0.01),
→ Please keep the consistent spacing in the expression of statistical value (e.g., p<0.01 or p <0.01).
Discussion: 4.5. Mortality and causes of graft failure in PALF group
The researchers analyzed that the survival outcomes of the PALF group were inferior compared the that of the BA group in KM survival analysis. However, In multivariate analysis, PALF etiology was not a significant factor for patient survival. Therefore, researchers need a more detailed explanation of the reasons for the two different statistical results to be more persuasive.
Reviewer 3 Report
I congratulate the authors on this retrospective study on Asian experience in paediatric liver transplantation and the efforts they made to treat pediatric patients with acute liver failure.
Here are some questions I would like to ask:
- What are the main contraindications to liver transplantation in paediatric patients with PALF? (brain damage, and so on)
- Could you please define your paediatric age range in Korea (< 18yold ?)
- At the end of references there are two more references numbered 1 and 2, why?
